# Direct Synthesis of Oxynitride Nanowires through Atmospheric Pressure Chemical Vapor Deposition

**DOI:** 10.3390/nano10122507

**Published:** 2020-12-14

**Authors:** Babak Adeli, Fariborz Taghipour

**Affiliations:** 1Department of Chemical and Biological Engineering, University of British Columbia, Vancouver, BC V6T1Z3, Canada; badeli@chbe.ubc.ca; 2Clean Energy Research Centre (CERC), University of British Columbia, Vancouver, BC V6T1Z3, Canada

**Keywords:** GaN:ZnO, nanowires, atmospheric pressure CVD, vapor–liquid–sold

## Abstract

Binary and ternary oxynitride solid alloys were studied extensively in the past decade due to their wide spectrum of applications, as well as their peculiar characteristics when compared to their bulk counterparts. Direct bottom-up synthesis of one-dimensional oxynitrides through solution-based routes cannot be realized because nitridation strategies are limited to high-temperature solid-state ammonolysis. Further, the facile fabrication of oxynitride thin films through vapor phase strategies has remained extremely challenging due to the low vapor pressure of gaseous building blocks at atmospheric pressure. Here, we present a direct and scalable catalytic vapor–liquid–solid epitaxy (VLSE) route for the fabrication of oxynitride solid solution nanowires from their oxide precursors through enhancing the local mass transfer flux of vapor deposition. For the model oxynitride material, we investigated the fabrication of gallium nitride and zinc oxide oxynitride solid solution (GaN:ZnO) thin film. GaN:ZnO nanowires were synthesized directly at atmospheric pressure, unlike the methods reported in the literature, which involved multiple-step processing and/or vacuum operating conditions. Moreover, the dimensions (i.e., diameters and length) of the synthesized nanowires were tailored within a wide range.

## 1. Introduction

Direct fabrication of oxynitride thin films through vapor phase strategies has remained extremely challenging and the reported studies in the open literature placed emphasis on multiple-step processing and/or vacuum operating conditions. Here, the local mass transfer flux of vapor deposition is enhanced through engineering the mass transfer mechanism and a direct and scalable catalytic VLSE route for the fabrication of oxynitride solid solution nanowires from their oxide precursors is proposed.

Among oxynitride semiconductors, gallium-zinc oxynitride solid solution (GaN:ZnO), also denoted as $\left({\mathrm{Ga}}_{1-x}{\mathrm{Zn}}_{x}\right)\left(N_{1-x}O_{x}\right)$, where *x* is the composition of ZnO in the host GaN lattice, is one of the most promising materials that exhibited outstanding stability and remarkable overall water splitting performance within the visible region [1,2,3]. In the past decade, various strategies, including structural [4,5,6,7] and surface [8,9,10,11] modifications as well as interfacial hybridization [12,13], have led to improvements in the photocatalytic activity of the GaN:ZnO photocatalyst. However, its quantum efficiencies do not exceed a few percent, as photo-generated charges in the bulk structures and close-packed crystallites tend to recombine at the structural defects and grain boundaries [14,15].

Synthesis GaN:ZnO solid alloy in one-dimensional morphology is practiced lately to minimize the rate of undesired charge carriers recombination. Han and co-workers employed vacuum chemical vapor deposition (CVD) and ZnGa_2_O_4_ nanoparticles as the source material for the synthesis of single crystal GaN:ZnO nanowires [16]. Modifications in epitaxial growth condition in recent years resulted in fabrication of oxynitride thin film [17,18], and 1-D heterostructures [19,20]. GaN:ZnO solid solution nanotubes were prepared through diffusion of the ZnO core into the GaN shell through epitaxial casting route [14]. Zhong et al. proposed a two-step template-base fabrication for highly ordered and dense nanowire arrays of ZnO-GaN:ZnO core-shell through high temperature vapor-phase diffusion of Ga and N into a single crystal ZnO nanowire lattice [21]. Recently, Li et al. developed a sol-gel solid state route for the fabrication of Zn-rich GaN:ZnO nanorods [22]. Here, the direct synthesis of GaN:ZnO nanowires through atmospheric pressure chemical vapour deposition is investigated. The synthesis of the gallium-zinc oxynitride thin film was carried out in a single-zone tube furnace, where the hydrodynamics of the vapor-liquid-solid epitaxy (VLSE) growth region were modified. To the author’s knowledge, direct synthesis of gallium-zinc oxynitride thin film at atmospheric pressure is not realized, and the methodology proposed in this study is the first example of facile and scalable synthesis of GaN:ZnO nanowires.

The gas mass-transfer flux ($J_{g}$) of vapor phase deposition is a function of the overall gas transfer coefficient ($k_{g}$) and the partial pressure difference$\;\left(P_{g}-P_{i}\right)$, which can be expressed as $J_{g}=k_{g}\cdot \left(P_{g}-P_{i}\right)$. To maintain stable growth, chemical vapor deposition (CVD) synthesis is conventionally operated in the laminar regime [23]. The partial pressure of precursors ($P_{g}$) is sufficiently high in the neighboring regions of the source materials; however, it decays substantially in the vicinity of the catalyst and the fluid–catalyst interface ($P_{i}$), due to the low mixing characteristics of the laminar flow, which are correlated to the thickness of the boundary layer. Further, as demonstrated in Figure 1a, the low velocity of fluid at the growth region creates a relatively thick concentration boundary layer (CBL), resulting in a low transfer coefficient ($k_{g}$) that is inversely correlated to the thickness of the boundary layer, significantly reducing the partial pressure of reactants at the fluid-catalyst interface and, accordingly, their equilibrium concentration (*C_i_*). Therefore, the catalyst–reactants’ alloy composition concentration (*C_l_*), which is the driving force of primary nucleation and growth, is substantially decreased. As a result, the gas diffusion flux of gaseous products becomes too low to support stable epitaxial growth, and synthesis of an oxynitride thin film is not realized.

To enhance the gas mass transfer flux at atmospheric pressure, the hydrodynamics of the CVD process must be controlled; as such, a higher concentration gradient profile is regulated at the growth region. By increasing the local velocity of the fluid, the mass transfer coefficient, which could be a function of both the molecular diffusion and the eddy diffusion or turbulence, improves significantly. Therefore, tailoring the CVD process hydrodynamics could result in sufficient concentration gradient near the catalyst, leads to a high mass transfer coefficient of the reactants; subsequently, direct fabrication of oxynitride nanowires at atmospheric conditions.

As illustrated in Figure 1b, the fluid crossing at the downstream of the source materials was reduced, thus increasing the fluid velocity. Therefore, the low vapor pressure of building blocks caused by the atmospheric pressure operating condition was compensated by an improved partial pressure (or concentration) gradient and the availability of reactants at the substrate region, leading to enhanced mass transfer at the gas-catalyst interface. Under such an operating condition, while the CVD process is carried out in a laminar flow, a local pseudo-turbulent regime at a microscopic level was developed near the catalyst surface, as illustrated in Figure 1c. At the same time, a high local vapor pressure gradient on the surface of the substrate was maintained, as demonstrated in Figure 1d, leading to a substantial enhancement in the mass transfer flux, and the subsequent formation of a feasible VLSE growth kinetics at atmospheric pressure. Pictures of the CVD apparatus and implemented modifications to enhance the local mass transfer flux of vapor deposition are presented in Appendix A.

## 2. Experimental Section

### 2.1. GaN:ZnO Nanowires Synthesis

Ga_2_O_3_ (99.99+%, Sigma-Aldrich, Oakville, ON, Canada) and ZnO (99+%, Sigma-Aldrich, Oakville, ON, Canada) were used as the source materials for the CVD growth of GaN:ZnO nanowires, as received and without further purifications. Ultra-pure Ar and NH_3_ gasses, polished quartz plates and Si (100) substrates were purchased from Praxair (Danbury, CT, USA), TGP (Painesville Township, OH, USA) and Ted Pella Inc. (Redding, CA, USA), respectively.

Deposition of the Au catalyst thin film was conducted in a high-vacuum Leica EM MED 020 coating system (Leica, Concord, ON, Canada) for planar magnetron sputtering, equipped with a quartz crystal film thickness monitor. Substrates were ultra-sonicated in a 50–50 V% isopropanol-ethanol solution, dried at room temperature and subjected to 10 nm Au coating.

The APCVD growth of the GaN:ZnO thin film was performed in a 40 mm quartz tube inside a 400 mm single-zone Nanyang Xinyu SK2-4-12TPB4 tube furnace (Nanyang City, Henan Province, China) with ±1 °C constant temperature control precision. The fluid crossing of the tube at the downstream of the source materials was reduced using an alumina filler. A uniform mixture of Ga_2_O_3_ and ZnO (1:1 wt) was loaded in an alumina boat and placed at the centre of the heated zone. The Au-coated substrates were secured at various locations in the quartz tube, as described in the text. The quartz tube was purged with ultra-pure Ar gas for 1 h, and then switched to ammonia 30 min prior to heating. The reaction tube was heated at 20 °C min^−1^ while NH_3_ gas was flowing at 250 mL min^−1^ at atmospheric pressure. After a given growth time, furnace heating terminated automatically, and the substrates were cooled at room temperature under ammonia flow. The synthesized thin films were washed with pure water and a 10 V% ethanol aqueous solution several times to ensure that their surfaces were free of ammonia residues.

### 2.2. GaN:ZnO Powder Synthesis

The layered double hydroxide (LDHs) of Ga^3+^ and Zn^2+^ was prepared trough the procedure described in the literature [5]. Ga_2_O_3_ (99.99+%, Sigma-Aldrich) and ZnO (99+%, Sigma-Aldrich) powders with [Zn]/[Zn + Ga] = 0.2 were dissolved in 8M HCl (Fisher Scientific, Waltham, MA, USA) at boiling temperature. The pH of the solution was adjusted to 9 with a mixed aqueous solution of 2M NaOH (≥97%, Fisher Scientific) and 1M Na_2_CO_3_ (≥99.50%, Fisher Scientific), added drop-wise to the acidic solution under vigorous stirring. The obtained slurry was aged in the mother liquor for 18–24 h at 80 °C. The precipitates were collected and washed thoroughly with pure water and ethanol, and subsequently dried overnight in a laboratory oven at 80 °C. The dried LDHs powder was mixed with Zn (99+%, Sigma-Aldrich) [Zn]/[Ga] = 0.25 and urea powders (Sigma-Aldrich) in a stainless-steel ball-mill jar for 10 min to obtain a uniform precursor. The prepared precursor was loaded into a semi-sealed alumina synthesis vessel and placed in a preheated laboratory muffle furnace for solid-state preparation at 900 °C for 12 min. The thermally treated sample was collected, refluxed in 10 V% ethanol aqueous solution at 60 °C for 4 h and washed with pure water. The washed sample was dried in the oven overnight and annealed in air at 600 °C for 1 h.

### 2.3. Characterization

To study the surface morphology of the thin film samples and their composition, an FEI Helios NanoLab 650 dual beam SEM (Milpitas, CA, USA) equipped with an EDAX TEAM Pegasus EDX spectrometer (EDAX, Hamilton, ON, Canada) was utilized. For imaging, the samples were coated with a 4–8 nm Ir layer using a high-vacuum Leica EM MED 020 sputter coater. The edges of the samples were painted with a silver suspension to ensure adequate conductivity. Imaging was conducted at 2–5 KeV applied voltage and 13–50 pA current. The average diameters and lengths of the nanowires were estimated using ImageJ 1.50i software, measuring the dimensions of 20 nanowires exhibiting the dominant morphology.

The synthesized nanowires were characterized based on their composition using EDX spectroscopy. At 5.0 KeV applied voltage, Ga Lα at 1.098 KeV, Zn Lα at 1.012 KeV, N Kα at 0.392 KeV, O Kα at 0.525 KeV, Au Mα at 2.123 KeV and Si Kα 1.740 KeV energy lines were used to identify and quantify the elements in the prepared sample [24]. Accordingly, Ga Kα (9.251 KeV) Zn Kα (8.637 KeV) energy lines were also used for elemental quantification at 20.0 KeV.

A Bruker D8 Advance XRD (Bruker, Madison, WI, USA) in Bragg–Brentano configuration with Cu Kα radiation was used to determine the crystalline phases and the structures of the prepared precursors and samples through XRD measurement. The analysis was terminated at 2θ = 65° due to the intense Si peak observed at 2θ~67°. The d-spacing of the crystallographic planes was calculated using the Bragg’s law Equation (1) [25]:(1)$$d_{\left(h,k,l\right)}=\frac{\lambda }{2\sin\left(\theta \right)}\;,$$
where h, k, and l are the crystal facet Miller indices, λ is the X-ray wavelength equal to 1.54 A and θ (degree) is the diffraction angle (2θ) divided by 2.

The synthesized photocatalyst crystalline structures were investigated through TEM and STEM using an FEI Tecnai Osiris (Hamilton, ON, Canada) S/TEM system equipped with an Analytical TWIN (A-TWIN) objective lens integrated with the Super-X EDX detection system, Gatan Ultrascan CCD camera, high-angle annular dark-field (HAADF), and BF STEM detectors, capable of STEM/EDX mapping (Esprit) and S/TEM tomography. The lattice spacing in the HRTEM images was estimated using ImageJ 1.50i software. For the TEM/STEM sample preparation, the coated Si substrate was covered with a few drops of ethanol, and its surface was scratched with a knife several times. The obtained suspension was ultra-sonicated for a given time and drop casted on a Ted Pella carbon Type-B, 200 mesh TEM grid.

The density of crystal defects and optical excitation band of the prepared samples were studied by photoluminescence (PL) intensity analysis acquired by the combined system of a Princeton Instruments SP2300i (Princeton Instruments, Trenton, NJ, USA) and a Hamamatsu C7700 streak camera (Hamamatsu, San Jose, CA, USA). The excitation source was an EKSPLA PL2241 picosecond laser (Ekspla, Vilnius, Lithuania) with 10 Hz repetition rate and 30 ps pulse duration at 355 nm.

The ANSYS 18.2 simulation package (Canonsburg, PA, USA) was used to build, discretize, and perform the computational Fluid Dynamics (CFD) simulation. The problem domains were discretized via 25813 hexahedral. Mesh size provided grid-independent results as further mesh refinement had a negligible impact on the final solutions. Velocity inlet boundary condition was applied to the inlet and evaporating substrate. Pressure-outlet and no-slip boundary conditions were imposed on the outlet and internal walls, respectively. The SIMPLE was selected for the pressure-velocity coupling. The second-order upwind was chosen for pressure, momentum, and species under spatial discretization. Finally, 0.8 was used for the species under-relaxation factor. The convergence of the results was approved once the errors of two successive iterations for the scaled residuals were less than $10^{-6}$.

## 3. Results and Discussion

The results of our experiments indicated the effectiveness of vertical substrate orientation, as displayed in Figure 1a, compared to horizontal face-up or face-down positions (SEM images of substrate treated at horizontal face-up and face down orientation are illustrated in Appendix A), which are attributed to the formation of a local thin boundary layer and uniform temperature profile in the growth region at the fluid-catalyst interface. The successful epitaxial synthesis of oxynitride nanowires was realized using Si (100) substrate, which was processed at a source temperature (T_s_) = 1000 °C with a growth time (t) = 30 min, controlled by the distance downstream of the centre of the heated zone (D) = 8 cm. The non-epitaxial growth through the proposed methodology was also investigated, and the results are presented in the Appendix A. The low-resolution scanning electron microscope (SEM) image of the synthesized nanowires in Figure 2a indicates the uniform coverage of the nanostructures at high density. The presence of the Au catalyst on the tip of almost all the nanowires in Figure 2c confirms their vapor–liquid–solid growth mechanism. As shown in Figure 2b, after CVD growth, the substrate was observed to have a matte finish on the region exposed to fluid, suggesting a homogenous centimetre-scale growth. Non-uniform growth of one-dimensional structures are observed at the regions close to the area used to secure the substrate on the alumina base, likely due to variations in hydrodynamics regime in this region, as displayed in Figure 1c. It is important to note that VLSE growth kinetics is a function of mass, momentum, and heat transfer; thus, the proposed technique offers enhanced mass and momentum transfer, while maintaining a uniform growth temperature, since the temperature is in equilibrium across the substrate.

Nanowires with an average diameter and length of d~190 nm and L~3.76 μm, respectively, were synthesized at D = 8 cm and T_s_ = 1000 °C for t = 30 min. Accordingly, nanowires with d~205 nm and L~1.15 μm were obtained by moving the location of the substrate to a farther distance (D = 13 cm). The average areal density of the nanowires was approximately ~4.1 × 10^7^ cm^−2^, favourably comparable to those reported for the thin films of GaN prepared via vacuum metal organic CVD (1.5 × 10^4^ cm^−2^) [26] and molecular beam epitaxy (1.0 × 10^10^ cm^−2^) [27]. Approximately 31% of the Au nanoparticles ($d_{p}\;$= 80–300 nm) were utilized for nanowire growth, slightly lower than the 49% reported for the dense ZnO nanowire arrays prepared via vacuum CVD [28]. Such a difference is likely attributable to the morphology of the prepared nanowires, which appear to be kinked, bent in multiple locations towards the growth direction, as demonstrated in Figure 2c. The overall morphology of the nanowires (ordered or kinked) does not necessarily determine their functionality for a particular application, unless it significantly affects the defect density and/or active surface area of the thin film. In fact, disoriented or kinked nanowires are believed to be more effective for (photo)-electrochemical applications, considering the hindered diffusion of the reactants within the ordered nanowire arrays and the efficient transfer of (photo)-electro-induced charges via structural defects. The origin of such a kinked morphology of nanowires has been attributed to the formation of a local temperature-pressure domain in the regions around an Au catalyst, as a nanowire grows faster along the direction with higher vapor pressure [29]. The microscopic pseudo-turbulency creates spectrums of temperature-pressure domain at the growth area, altering the VLSE growth kinetics towards various crystallographic planes. Further, based on the Wulff theorem [30], the nanowire growth direction could have changed in order to maintain the minimum surface energy of the nanowires’ structure. In such a condition, the Au catalyst nanoparticles continuously migrate from smaller droplets to the larger ones (Ostwald ripening), leading to variations in the length, shape and sidewall properties of the nanowires [31]. The relatively large difference between the size of Au catalyst and diameter of the nanowire displayed in the inset of Figure 2c suggests a self-regulating VLS mechanism that controls the shape and size of the Au nanoparticles, as well as the exposed crystalline facets.

The compositional data obtained from energy dispersive X-ray spectroscopy indicate the formation of the GaN:ZnO solid solution phase with ~20% deviation (Zn/O = 0.78 and Ga/N = 1.23) from the stoichiometric ratio $\left({\mathrm{Ga}}_{1-x}{\mathrm{Zn}}_{x}\right)\left(N_{1-x}O_{x}\right)$, with *x* = [Zn]/[Ga] + [Zn] = 0.10 (Zn/Ga = 0.11). The deviation from stoichiometric values is most likely related to (1) surface absorption of oxygen from ambient, which increases [O] and reduces the [Zn]/[O] ratio from its stoichiometric value of 1.0, (2) high temperature evaporation of Zn, which reduces the [Zn]/[O] ratio from its stoichiometric value of 1.0 [3], (3) partial collapse of Ga–N and Zn–N bonds at high temperature due to oxidation that increases the [Ga]/[N] and decreases the [Zn]/[O] ratios from their stoichiometric value of 1.0, and (4) incomplete solid-state nitridation due to low synthesis time which results in high and low composition of oxygen and nitrogen, respectively; thereby reducing the [Zn]/[O] and enhancing the [Ga]/[N] ratios.

Since the VLSE growth process is controlled by the partial pressure of building blocks, the ratio of Ga_2_O_3_/ZnO plays an important rule in the final characteristics of the fabricated GaN:ZnO nanowires. Our screening study on the effect of Ga_2_O_3_/ZnO ratio on VLSE growth of GaN:ZnO nanowires suggest that higher concentration of ZnO in precursor, beyond Ga_2_O_3_/ZnO = 1 wt, does not increase the Zn/Ga ratio in the thin film alloy. This is likely attributed to the hindered condensation of Zn at high growth temperature [3]. In some studies, more kinked morphology was observed for the thin films which were prepared with ZnO/Ga_2_O_3_ > 1 wt, although no definitive trend could be established.

The elemental profiles displayed in Figure 3a are indicative of the presence of the Au catalyst at the cross-section of the nanowire tip, as well as the highly uniform distribution of core elements along the growth axis, with no sign of drastic changes in the nanowire composition. The low Zn content of the solid solution is attributable to the high growth temperature at D = 8 cm, which is suitable for the VLS synthesis of GaN nanowires, but relatively greater than that of ZnO nanowires. There was a low Au composition on the surface of the nanowires, where the Au catalyst droplets wetted the nanowire sidewalls during VLS growth.

The hexagonal crystalline structure of the prepared GaN:ZnO nanowires can be observed from the X-ray diffraction (XRD) pattern in Figure 3b. The high intensity of the (002) peak in the XRD pattern, in contrast to its bulk counterpart [32], testifies to the 1-D epitaxial growth, primarily towards the hexagonal *c*-axis, with the estimated lattice spacing of 0.26 nm. The (101) peak is attributed to the secondary growth direction, which is most likely responsible for the kinked shape of the prepared nanowires, as well as six symmetrical crystallographic planes parallel to the [002] axis. The favourable growth direction and exposed facets are denoted in the scanning transmission electron microscopy (STEM) image presented in the Appendix A.

The small peak at 2θ~32.5° is most likely related to partially crystalized GaZnO compound, as also observed in solid-state synthesis of GaN:ZnO powder via amonolysis of Ga^3+^/Zn^2+^ layered double hydroxides [5]. This peak is expected to decline when VLSE growth is carried out at higher temperature. This would be challenging to accomplished as higher temperature promotes the faster evaporation of Zn, which is undesired. Because of the low intensity of this peak and considering the nature of XRD analysis as a bulk characterization method, it seems this peak does not indicate the presence of a crystalline structure of GaZnO within the nanowires lattice, rather suggests the surface deposition of this compound throughout the VLSE growth.

The high-resolution STEM image in Figure 3c reflects a single crystal structure, consisting of well-ordered parallel lattice fringes, with no sign of defects. In agreement with the XRD data, the lattice spacing towards the growth direction was estimated to be ~0.26, inferring the [002] one-dimensional (1-D) crystallographic growth. This observation is also confirmed by the selected area electron diffraction pattern (SAED) of the nanowires in the inset of Figure 3c, exhibiting a (002) plane diffraction pattern.

The hexagonal cross-section of the nanowire in Figure 3d indicates the *c*-axis as the primary VLSE crystallization. Different crystallographic facets have been observed on the lateral surfaces of some nanowires. Compared to the nanowires with dominant exposed (10$\bar{1}$0) facets, a portion of the nanowires exhibited lateral walls consisting of (10$\bar{1}$1) and (10$\bar{1}\bar{1}$) planes, evidenced by the ~120° angle between the two. The transformation of the sidewall facets from the typical (10$\bar{1}$0) to polarized (10$\bar{1}$1) and (10$\bar{1}\bar{1}$) is related to the self-regulating kinetics of the VLS mechanism, as described earlier. It is highly desirable to control the surface polarity of nanowires because different crystallographic facets exhibit vastly different reactivity for particular catalytic reactions [33,34].

The photoluminescence (PL) analysis of a the prepared GaN:ZnO nanowires which is compared with the non-stoichiometry GaN:ZnO solid solution synthesized through amonolysis of Ga^3+^/Zn^2+^ layered double hydroxides (illustrated in Appendix A), suggesting low density of the crystal defects for the 1-D crystallites. The PL peak for the nanowires sample is observed at a lower wavelength, compared to powder GaN:ZnO. This shift is likely attributed to the lower Zn-content of nanowires, resulted from high temperature Zn evaporation at the longer synthesis time.

For 1-D nanostructures, controlling the morphology of the synthesized thin film (i.e., diameter and length) is known to have a significant impact on the performance of the functional materials. However, dimension control has not been addressed for the majority of the fabrication routes reported in the literature. As suggested by our experiments, due to the low concentration of building blocks, the source and growth temperatures had significant effects on the morphology of the synthesized thin films; thus, small variations in either could alter the VLSE kinetics. To study the mechanisms that control the morphology of the thin films, the oxynitride nanowires were prepared at various source temperatures and durations (T_s_ = 900, 1000, and 1100 °C for t = 5, 30 and 60 min, respectively), as well as distances from the centre of the furnace (0, 8 and 13 cm), which influence the substrate temperature. The dimensions and SEM images of the synthesized nanowires are presented in Figure 4 and in the Appendix A, respectively. Although higher CVD growth temperature (function of T_s_ and D) reduces the gaseous products’ condensation, the experimental data suggest the significant effect of local vapor pressure of building blocks, which was greater at the regions approaching the centre of the furnace. Thus, the synthesis of longer nanowires was realized at higher growth temperatures. This phenomenon explains the longer nanowires at T_s_ = 900 °C/D = 0 cm, T_s_ = 1000 °C/D = 0 cm, T_s_ = 1000 °C/D = 8 cm, and T_s_ = 1100 °C/D = 8 cm in Figure 4a, compared to those prepared at D = 13 cm (lower T_g_). For instance, the length of the nanowires synthesized varies from 0.52 μm to 4.51 μm when D changes from 13 cm to 0 cm at T_s_ = 900 °C, as illustrated in Figure 4a. A similar trend can be observed for T_s_ = 1000 °C (L = 1.15 μm and 4.39 μm for D = 13 cm and 0 cm, respectively) and T_s_ = 1100 °C (L = 1.26 μm and 4.03 μm for D = 13 cm and 8 cm, respectively). To obtain nanowires with larger aspect ratios (or smaller diameters), the source temperature must be sufficiently high to support a high local vapor pressure, hence, causing greater mass transfer flux. Therefore, regardless of the growth temperature (or distance between substrate and centre of furnace), the synthesis of thin nanowires (large aspect ratios) cannot be realized at relatively low source temperatures, as can be seen for T_s_ = 900 °C in Figure 4b. The relatively large variations in the nanowires’ dimension at D = 0 is attributed to the concentration-turbulent profile in this region. It can be concluded that the diameter of the nanowires is a function of T_g_ and T_s_, and as such, the availability of the building blocks is controlled by T_s_, while the rate of aggregation of Au droplets is regulated by T_g_. The outcome of these competing processes results in the fabrication of GaN:ZnO thin films consisting of various morphologies, with average diameters ranging from 105 to 340 nm and average lengths from 0.52 to 4.51 μm, as shown in the Appendix A.

As displayed in Figure 4b, the Zn content of the prepared thin film can be controlled in regard to the growth temperature, and a uniform thin film consisting of GaN:ZnO nanowires with Zn/Ga ratios up to 0.12 (*x* = 0.11) were synthesized, likely the highest Zn content that can be achieved at atmospheric condition, where the Zn content of the nanowires increases when T_g_ decreases. Further adjustments to the current apparatus, such as placement of ZnO source at the downstream of the Ga_2_O_3_ precursor, would enable the synthesis of Zn-rich nanowires, with considerable red-shift in absorption spectrum, and a subsequently superior response to visible light [5].

## 4. Conclusions

The direct synthesis of GaN:ZnO nanowires is realized through atmospheric pressure chemical vapour deposition. The hydrodynamics of the vapor-liquid-solid epitaxy in the growth region of a single-zone tube furnace were modified, which enhanced the concentration gradient profile of oxynitride building blocks at the growth region, resulted in enhanced concentration gradient near the catalyst, and subsequently, direct fabrication of oxynitride nanowires at atmospheric conditions. Pseudo-uniform coverage of oxynitride nanowires in centimetre-scale on Si (100) substrate with ~4.1 × 10^7^ cm^−2^ areal density was confirmed through SEM micrographs. The compositional data obtained from X-ray spectroscopy indicated the formation of the GaN:ZnO solid solution phase with *x* = [Zn]/[Ga] + [Zn] = 0.10 (Zn/Ga = 0.11). XRD and HRTEM data testified to the 1-D epitaxial growth, primarily towards the hexagonal *c*-axis, with the estimated lattice spacing of 0.26 nm. Finally, the morphology of the synthesized thin film were controlled over a wide range, and formation of GaN:ZnO nanowires with average diameters ranging from 105 to 340 nm and average lengths from 0.52 to 4.51 μm accomplished. The direct VLSE methodology proposed here can be readily adapted for the fabrication of wide spectrum of binary and ternary solid alloy thin films.

## Figures and Tables

**Figure 1 nanomaterials-10-02507-f001:**
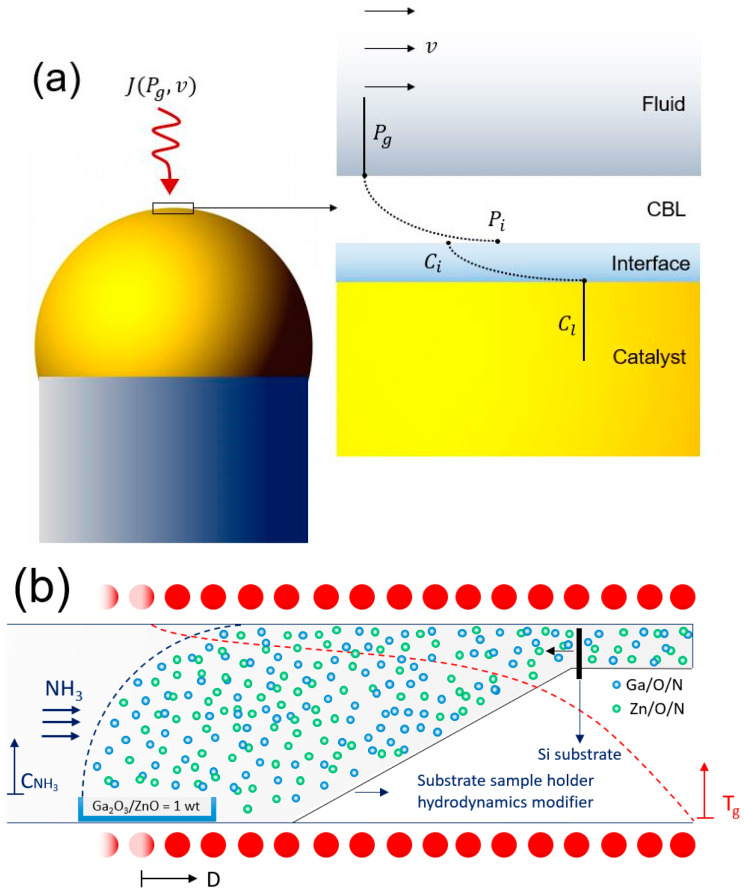
Controlling the hydrodynamics of the CVD process towards atmospheric chemical vapor deposition of nanowires. (**a**) Schematic illustration of diffusion of a compound from fluid with partial pressure of P_g_ through a CBL with partial pressure of P_i_, fluid-catalyst interface with partial composition of C_i_, to catalyst droplet with C_l_ partial composition. The effective parameters on the interfacial gas–liquid mass transfer flux (*J*), including fluid velocity (*v*), are shown. (**b**) CVD synthesis of nanowires: an alumina boat loaded with a mixture of source materials (1:1 wt of Ga_2_O_3_ and ZnO) was placed at the centre of a horizontal tube furnace at source temperature (T_s_). The furnace geometry downstream was modified using an alumina filler. The Si (100) substrate was secured at growth temperature (T_g_) in a vertical position at a distance from the centre of the heated zone (D) downstream of the source materials. The schematic temperature profile of the tube furnace, concentration of ammonia (NH_3_) carrier/reactant gas and schematic Ga and Zn atoms’ building blocks are shown. (**c**) Velocity profile of the gaseous reactants at the vicinity of Au-coated substrate calculated by computational fluid dynamics, showing high local velocity near the surface of the substrate, which enhances the concentration gradient and, subsequently, the local mass transfer coefficient (the length and colour of the vectors represent the velocity magnitude). (**d**) Mass fraction profile of the mix-metal oxynitride building blocks, calculated by computational fluid dynamics, showing a high local concentration gradient on the surface of the substrate.

**Figure 2 nanomaterials-10-02507-f002:**
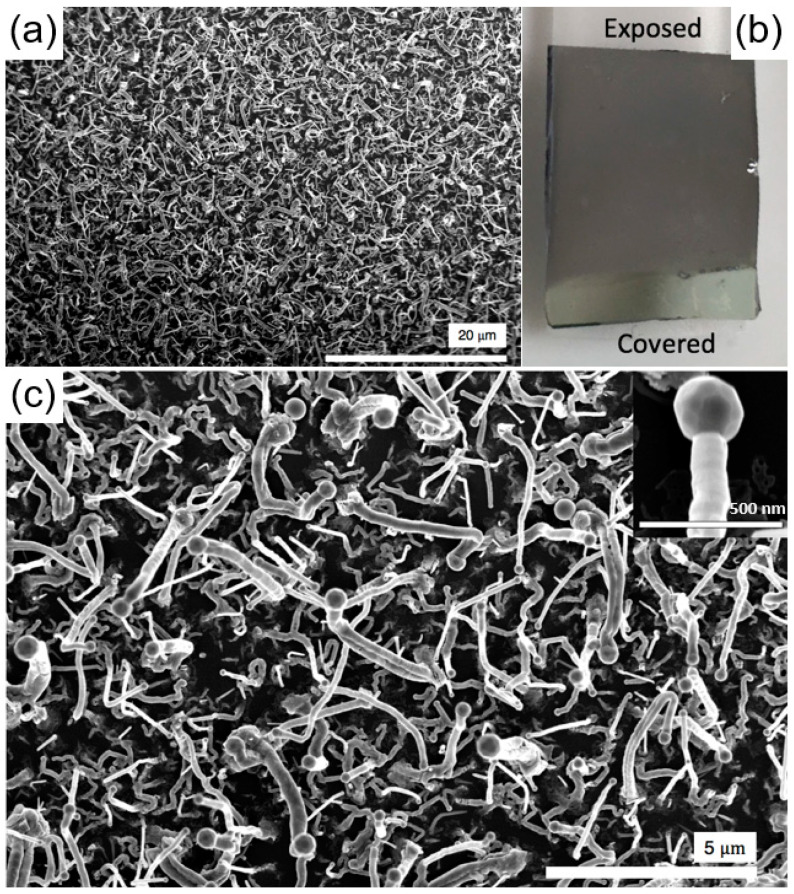
Morphology of the synthesized nanowires. (**a**) Low-resolution SEM image of the nanowires on the substrate. (**b**) Digital photo of a coated Si substrate from the region exposed to the reactants. (**c**) High-resolution SEM image of the nanowires on the substrate. Inset shows an individual Au-capped nanowire—Scale bar = 500 nm. Nanowires were prepared at Ts = 1000 °C for 30 min at D = 8 cm downstream of the Ga_2_O_3_ and ZnO source materials.

**Figure 3 nanomaterials-10-02507-f003:**
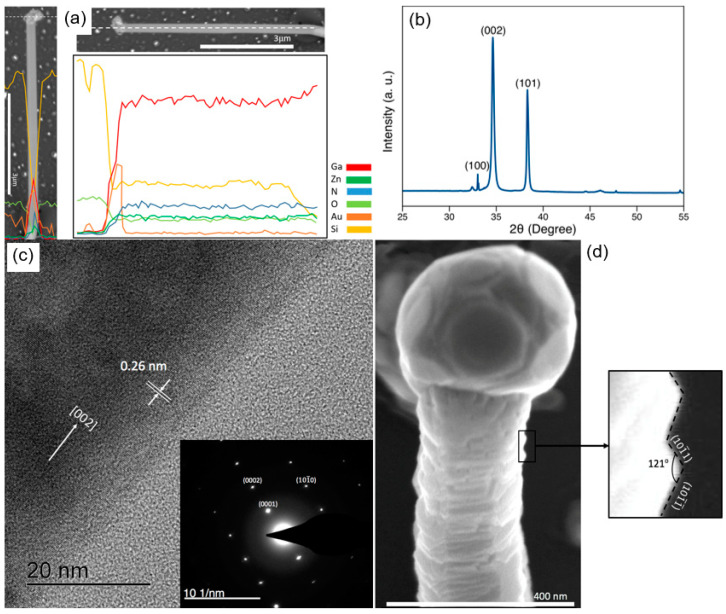
Chemical and structural characteristics of the synthesized nanowires. (**a**) Elemental profile of an individual nanowire along the tip cross-section and growth direction. (**b**) XRD pattern of the prepared thin film. (**c**) HRTEM images of an individual GaN:ZnO nanowire, showing the distance between two parallel crystal fringes and the favourable growth direction of the nanowires. Inset displays the SAED pattern of the nanowire. (**d**) High-resolution SEM image of an individual nanowire, showing the exposed (10$\bar{1}$1) and (10$\bar{1}\bar{1}$) facets, determined from their position with respect to the [002] and the angle between the planes. The nanowires were prepared at T_s_ = 1000 °C for 30 min at D = 8 cm downstream of the Ga_2_O_3_ and ZnO source materials.

**Figure 4 nanomaterials-10-02507-f004:**
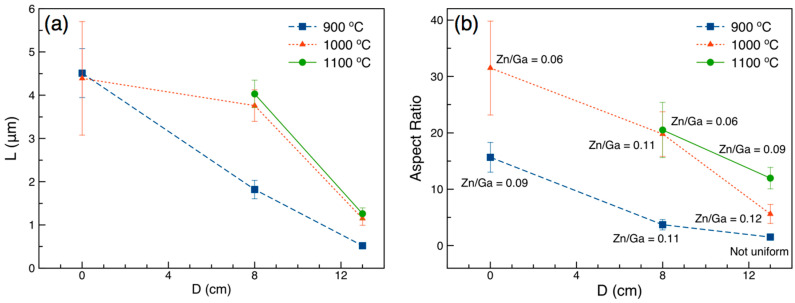
Dimensions of synthesized GaN:ZnO nanowires. (**a**) Length and (**b**) aspect ratios of the nanowires prepared at various source temperatures, set to 900 °C to 1100 °C, and locations (D) downstream of the Ga_2_O_3_ and ZnO source materials. Error bars represent a 90% confidence interval. Uniform coverage was not realized at T_s_ = 900 °C for 60 min at D = 13 cm; thus, the results are presented based on a few available nanowires observed in the study region.

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
