# Peer review of "Direct Synthesis of Oxynitride Nanowires through Atmospheric Pressure Chemical Vapor Deposition"

_nanomaterials, 2020, doi:10.3390/nano10122507_

Round 1

Reviewer 1 Report

The manuscript by Adeli et al. demonstrate the CVD synthesis of GaN:ZnO nanowires at atmospheric pressure. The synthesized nanowires were well characterized using SEM, STEM, XRD, and PL. From the perspective of the quality of the synthesized nanowires, the topic of this study is timely, and the review is well written in a scholarly manner. Therefore, I recommend this manuscript to be published in nanomaterials. Here is one question.

How does the ratio of the Ga2O3/ZnO precursor and amount of precursors affect the overall growth of the GaN:ZnO nanowires?

Author Response

RESPONSE TO THE REVIERS’ COMMENTS

Manuscript ID: nanomaterials-1016819

Reviewer #1

We appreciate the reviewer’s time in evaluating our manuscript and we are certainly thankful for his/her comments. Reviewer’s comments are addressed below:

Reviewer’s comment:

The manuscript by Adeli et al. demonstrate the CVD synthesis of GaN:ZnO nanowires at atmospheric pressure. The synthesized nanowires were well characterized using SEM, STEM, XRD, and PL. From the perspective of the quality of the synthesized nanowires, the topic of this study is timely, and the review is well written in a scholarly manner. Therefore, I recommend this manuscript to be published in nanomaterials. Here is one question.

Question 1:

How does the ratio of the Ga2O3/ZnO precursor and amount of precursors affect the overall growth of the GaN:ZnO nanowires?

Response to reviewer’s Question 1:

We thank the reviewer for raising this important question. Since the VLSE growth process is controlled by the partial pressure of building blocks, the ratio of Ga2O3/ZnO plays an important rule in the final characteristics of the fabricated GaN:ZnO nanowires. Due to peculiar characteristics of visible-light activated GaN:ZnO, an emphasis is placed on the synthesis of Zn-rich GaN:ZnO [1], because the Zn-content of GaN:ZnO is responsible for visible-light activity of the solid solution material. Similarly, the objective of our study focused on the synthesis of GaN:ZnO nanowires with high Zn-content. Our screening study on the effect of Ga2O3/ZnO ratio on VLSE growth of GaN:ZnO nanowires suggest that higher concentration of ZnO in precursor, beyond Ga2O3/ZnO = 1wt,  does not increase the Zn/Ga ratio in the thin film alloy. This is likely attributed to the hindered condensation of Zn at high growth temperature [1]. In some studies, more kinked morphology was observed for the thin films which were prepared with ZnO/Ga2O3 > 1 wt, although no definitive trend could be established.

To address the reviewer’s comments, the following text was added to the revised manuscript”

Since the VLSE growth process is controlled by the partial pressure of building blocks, the ratio of Ga2O3/ZnO plays an important rule in the final characteristics of the fabricated GaN:ZnO nanowires. Our screening study on the effect of Ga2O3/ZnO ratio on VLSE growth of GaN:ZnO nanowires suggest that higher concentration of ZnO in precursor, beyond Ga2O3/ZnO = 1wt,  does not increase the Zn/Ga ratio in the thin film alloy. This is likely attributed to the hindered condensation of Zn at high growth temperature [1]. In some studies, more kinked morphology was observed for the thin films which were prepared with ZnO/Ga2O3 > 1 wt, although no definitive trend could be established. [Page 14, Line 14–21]

Reviewer 2 Report

This paper describes the synthesis of GaN:ZnO nanowires using APCVD. The authors studied the growth of nanowire using an original system by modifying the hydrodynamics of the vapour phase on the surface of the substrate. They studied the GaN:ZnO nanowires by XRD, SEM, TEM and PL measurement. The paper is well written and fits with the scopes of nanomaterials. However, I recommend revisions before the publication of this manuscript.
Here are the detailed comments:

- The authors should add the number of the supporting information in the main manuscript.
- All scale bar legend should be visible on SEM and TEM images (inset fig 2c, S.I.6).
- Figure 2c show a large variation of nanowire diameter. The authors should add a diameter distribution diagram, similarly to what is done for nanoparticle size distribution diagram.
- In figure 3a, why Zn element is detected before the nanowire (yellow line).
- What is the XRD peak at 32.5degree; it is not an artefact.
- During the synthesis, the authors used NH3 as precursors. The EDX analyses indicate a ration Zn/O = 0.78 and Ga/N = 1.23. Which means an excess of nitrogen and a large deficit of oxygen. The error bars do not cover these differences compared to normal stoichiometry. It is known that 3 Zn + 2 NH3 induces the formation of Zn3N2 + 3 H2 at 600C. The nanowires certainly Zn3N2 as secondary phase and this should be discussed in the manuscript. The nanowire growth was performed at 1000C and 1100C.

Author Response

RESPONSE TO THE REVIERS’ COMMENTS

Manuscript ID: nanomaterials-1016819

Reviewer #2

We appreciate the reviewer’s detailed and thorough evaluation of our article and the constructive comments on the manuscript. We have addressed each comment either by making the necessary modifications throughout the manuscript or responding directly to his/her comments. Our response to each individual comment/question is discussed below:

Reviewer’s comment:

This paper describes the synthesis of GaN:ZnO nanowires using APCVD. The authors studied the growth of nanowire using an original system by modifying the hydrodynamics of the vapour phase on the surface of the substrate. They studied the GaN:ZnO nanowires by XRD, SEM, TEM and PL measurement. The paper is well written and fits with the scopes of nanomaterials. However, I recommend revisions before the publication of this manuscript.
Question 1:

The authors should add the number of the supporting information in the main manuscript.

Response to reviewer’s Question 1:

We thank the reviewer for the comment.  The numbers are provided in the revised manuscript.

Question 2:

All scale bar legend should be visible on SEM and TEM images (inset fig 2c, S.I.6).

Response to reviewer’s Question 2:

The scale bars for the inset of Figure 2.c and S.I.6 are revised per reviewer’s comment.

Question 3:

Figure 2c show a large variation of nanowire diameter. The authors should add a diameter distribution diagram, similarly to what is done for nanoparticle size distribution diagram.
Response to reviewer’s Question 2:

The range of average diameters of the prepared GaN:ZnO nanowires in various synthesis conditions is indicated between 105nm to 340 nm [Page 20, Line 3–4]. Also, Figure 4 of the revised manuscript depicts the effect of synthesis conditions on the length and aspect ratio of the nanowires, which accordingly reflects the effect of synthesis conditions on the overall morphology of the as synthesized nanowires in various growth conditions.

Question 4:

In figure 3a, why Zn element is detected before the nanowire (yellow line).
Response to reviewer’s Question 2:

We would like to request reviewer’s attention to the point that the yellow line in Figure 3a is related to Si substrate (not Zn), which is expected to exhibit higher concentrations in the region where nanowire composition is not scanned. On the other hand, the Zn concentration, which is displayed as green line in Figure 3a, shows increase in concentration in the regions GaN:ZnO nanowires is being scanned.

Question 5:

What is the XRD peak at 32.5degree; it is not an artefact.
Response to reviewer’s Question 2:

The small peak at 2θ ~ 32.5° is most likely related to partially crystalized GaZnO compound, as also observed in solid-state synthesis of GaN:ZnO powder via amonolysis of Ga3+/Zn2+ layered double hydroxides [1]. This peak is expected to decline when VLSE growth is carried out at higher temperature. This would be challenging to accomplished as higher temperature also promotes the faster evaporation of Zn, which is undesired.

To the authors opinion, because of the low intensity of this peak and considering the nature of XRD analysis as a bulk characterization method, this peak does not indicates the presence of a crystalline structure of GaZnO within the nanowires lattice, rather suggests the surface deposition of this compound throughout the VLSE growth.

To address the reviewer’s comment, the following text is added to the revised manuscript:

The small peak at 2θ ~ 32.5° is most likely related to partially crystalized GaZnO compound, as also observed in solid-state synthesis of GaN:ZnO powder via amonolysis of Ga3+/Zn2+ layered double hydroxides [1]. This peak is expected to decline when VLSE growth is carried out at higher temperature. This would be challenging to accomplished as higher temperature promotes the faster evaporation of Zn, which is undesired. Because of the low intensity of this peak and considering the nature of XRD analysis as a bulk characterization method, it seems this peak does not indicate the presence of a crystalline structure of GaZnO within the nanowires lattice, rather suggests the surface deposition of this compound throughout the VLSE growth. [Page 16, Line 13–20]

Question 6:

During the synthesis, the authors used NH3 as precursors. The EDX analyses indicate a ration Zn/O = 0.78 and Ga/N = 1.23. Which means an excess of nitrogen and a large deficit of oxygen. The error bars do not cover these differences compared to normal stoichiometry. It is known that 3 Zn + 2 NH3 induces the formation of Zn3N2 + 3 H2 at 600C. The nanowires certainly Zn3N2 as secondary phase and this should be discussed in the manuscript. The nanowire growth was performed at 1000C and 1100C.
Response to reviewer’s Question 2:

We thank the reviewer for pointing out to this important discussion.

It is known that Zn3N2 phase is stable at lower temperatures, since it has been observed that the Zn3N2 phase grew only in the vapor phase up to 463 °C [2]. For this reason, we find it unlikely that Z3N2 phase could be formed at temperature range employed in this study. To the authors opinion, the deviation from stoichiometric values is most likely related to:

  • Surface absorption of oxygen from ambient during the post treatment steps, which increases [O] and reduces the [Zn]/[O] ratio from its stoichiometric value of 1.0,
  • High temperature evaporation of Zn, which reduces the [Zn]/[O] ratio from its stoichiometric value of 10 [3],
  • Partial collapse of Ga–N and Zn–N bonds at high temperature due to oxidation which increases the [Ga]/[N], as well as decreases the [Zn]/[O] ratios from their stoichiometric value of 1.0, and
  • Incomplete solid-state nitridation due to low synthesis temperature which results in high, and low composition of oxygen and nitrogen respectively; therefore, reduces the [Zn]/[O] and enhances the [Ga]/[N] ratios.

To address the reviewer’s comment, the following text is added to the revised manuscript:

The deviation from stoichiometric values is most likely related to 1) surface absorption of oxygen from ambient, which increases [O] and reduces the [Zn]/[O] ratio from its stoichiometric value of 1.0, 2) high temperature evaporation of Zn, which reduces the [Zn]/[O] ratio from its stoichiometric value of 1.0 [3], 3) partial collapse of Ga–N and Zn–N bonds at high temperature due to oxidation that increases the [Ga]/[N] and decreases the [Zn]/[O] ratios from their stoichiometric value of 1.0, and 4) incomplete solid-state nitridation due to low synthesis time which results in high and low composition of oxygen and nitrogen, respectively; thereby reducing the [Zn]/[O] and enhancing the [Ga]/[N] ratios. [Page 14, Line 6–13]

References:

[1]        B. Adeli, F. Taghipour, Appl. Catal. A Gen. 521 (2016) 250–258.

[2]        P.C. Quayle, G.T. Junno, K. He, E.W. Blanton, J. Shan, K. Kash, Phys. Status Solidi 254 (2017) 1600718.

[3]        K. Maeda, K. Teramura, T. Takata, M. Hara, N. Saito, K. Toda, Y. Inoue, H. Kobayashi, K. Domen, J. Phys. Chem. B 109 (2005) 20504–20510.

Round 2

Reviewer 2 Report

The paper has been improved according to the suggested revisions. The novelty of this study is clearly defined and the presented work fits the requirements for publication in Nanomaterials. I have had a careful look at the authors’ responses and at the modifications performed in the manuscript. The authors performed all the necessary changes to improve their paper. I am satisfied of the changes made, and in my opinion the paper can be accepted for publication in Nanomaterials without further modifications.